# Sustainable Vegetable Oil-Based Biomaterials: Synthesis and Biomedical Applications

**DOI:** 10.3390/ijms24032153

**Published:** 2023-01-21

**Authors:** Chiara Nurchi, Silvia Buonvino, Ilaria Arciero, Sonia Melino

**Affiliations:** Department of Chemical Sciences and Technology—University of Rome Tor Vergata, Via della Ricerca Scientifica 1, 00133 Rome, Italy

**Keywords:** photopolymerization, biopolymers, 3D printing, tissue regeneration, lipids, scaffolds

## Abstract

One of the main criteria for ecological sustainability is that the materials produced for common use are green. This can include the use of biomaterials and materials that are environmentally friendly, biodegradable and produced at low cost. The exploration of natural resources as sustainable precursors leads to the production of biopolymers that are useful for 3D printing technology. Recently, waste vegetable oils have been found to be a good alternative source for the production of biopolymers in various applications from the engineering to the biomedicine. In this review, the processes for the synthesis of vegetable oil-based biomaterials are described in detail. Moreover, the functionalization strategies to improve the mechanical properties of these materials and the cell-material interaction for their potential use as micro-structured scaffolds in regenerative medicine are discussed.

## 1. Introduction

Polymers suitable for use in the biomedical field must be biocompatible, eventually biodegradable and non-cytotoxic. Natural polymers are generally sustainable, biocompatible and biodegradable materials, but difficult to purify and to use in 3D printing technology, and there is limited possibility for modulating their chemical-physical and mechanical properties. Oftentimes, the stiffness of the natural biopolymers is not tunable and, in many cases, they are also rapidly biodegraded. Alternatively, there are synthetic polymers, for which the de novo synthesis allows greater reproducibility and greater control of the chemical-physical and mechanical properties, but which are usually cytotoxic and require very laborious biocompatibility studies. 

Among the most commonly biocompatible polymers used in 3D printing technology, there are polymers derived from cellulose, polylactic acid, polycaprolactone, poly(lactic-co-glycolic) acid and polyethylenglycol (Figure 1).

Cellulose-derived polymers, mainly cellulose acetate and hydroxypropyl methylcellulose, are already widely used in the production of traditional pills and polypills capable of releasing multiple types of drugs in a controlled manner [1,2]. Nanocellulose can be deposited as a gel and printed obtaining three-dimensional structures that are transparent, thin and strong, which are suitable for use in wound-dressing, and able to create an optimal environment for wound healing, as they show antibacterial properties [3].

Polylactic acid (PLA) (Figure 1D) is an aliphatic polyester formed by the polymerization of lactic acid monomers, which can also be extracted from renewable sources such as corn starch or waste from the agri-food chain [4,5,6,7,8,9]. It is a biocompatible, biodegradable polymer, degradable by hydrolysis, with good printability, good thermal stability and good mechanical properties, which allow its use in the biomedical field [10]. Due to these characteristics, PLA is employed for the production of scaffolds of various shapes, sizes and porosities for use in musculoskeletal tissue engineering [11], for the regeneration of cartilage tissue [12] or for use as supports for the growth of bone tissue when combined with ceramic materials, such as calcium phosphate which is added to improve the material’s resistance [13]. However, PLA hydrolytically degrades in the body to form lactic acid that can induce inflammatory responses [14,15,16].

Polycaprolactone (PCL) (Figure 1E) is a semi-crystalline thermoplastic polymer obtained by ring opening of the cyclic monomer ɛ-caprolactone. It is biocompatible, biodegradable and presents good elasticity, stiffness and tensile strength [11] with a good hydrophilicity showing a very good potential for biomedical applications [17]. Scaffolds which are 3D printed from PCL and hydroxyapatite scaffolds possessing good porosity, a rough surface and bone-like firmness have been shown to be useful in bone regeneration, due to their ability to promote cell growth [18]. 

Poly(lactic-co-glycolic) acid (PLGA) (Figure 1F) is a copolymer of polylactic and polyglycolic acid (PGA) whose ester linkage hydrolysis in biological systems produces lactic acid and glycolic acid as metabolites, which are easily eliminated [19], but similarly to the PLA, can also produce a local inflammatory response. 

Polyethylene glycol (PEG) (Figure 1G) is a hydrophilic molecule characterized by hydroxylic groups that can be easily functionalized, e.g., carboxylate, thiolate, acrylate, etc., in order to promote both the photopolymerization process and the binding of bioactive molecules [20]. PEG-based materials show an easily tunable stiffness, are biocompatible and not immunogenic, and due to their specific properties, they are useful for photopolymerizable cellular scaffolds, implants and for the production of drug-releasing systems in biomedicine [21,22,23,24,25,26,27,28,29].

Thanks to their cytocompatibility, biocompatibility and biodegradability, PLA, PCL and PEG have received the approval of the FDA (Food and Drug Administration) for use in the fabrication of several medical devices such as sutures, grafts or micro/nanoparticles, and therefore, in recent years, they have been widely used in the biomedical field. 

Research interests are currently moving towards the production of new sustainable and recyclable materials and towards the improvement of traditional production processes for the waste valorization. In this context, biopolymers derived from vegetable oils are obtained from renewable sources and represent a good alternative to polymers derived from petroleum, as they have similar structures to monomers obtained from the oil industry [2,30,31,32,33,34,35,36,37,38,39,40,41]. Therefore, waste oils derived from the agri-food industry can be used for the production of both biofuel by transesterification [42] and biopolymers. Oil-based biomaterials also show great potential in 4D printing technology. In this technique, the fourth dimension is the time and the material confers to the 3D printed objects the ability to change form and function after 3D printing, offering additional capabilities and performance [43,44]. In this review, the synthesis of the vegetable oil-based biomaterials and their potential applications as scaffolds in regenerative medicine [45] are described in detail. 

## 2. Biopolymers Based on Vegetable Oils

### 2.1. Synthesis of Vegetable Oil-Based Polymers

Vegetable oils, mainly obtained from fruits and oilseeds, are vegetable fats which are liquid at room temperature consisting of a mixture of lipids, particularly of triglycerides (Figure 2). The most common vegetable oils are olive, soybean, peanut, sunflower, flax, corn and canola, and they are used in condiments, as frying oils and in the production of soaps, paints, oil paints and biofuels. Vegetable oils have several reactive sites for their functionalization, such as the ester groups and the double bonds found on unsaturated fatty acid that can be chemically modified by acrylation, transesterification, metathesis and epoxidation reactions, which allow the triglycerides to be transformed into polymerizable monomers [46]. Epoxidation is a favorite reaction in organic chemistry because the epoxy rings are excellent reaction intermediates that can be transformed into a large variety of functional groups. The oxidation of the double bonds present in the triglycerides with molecules such as hydrogen peroxide (H_2_O_2_) or peracids leads to epoxidized vegetable oils (EVOs), which are useful in the synthesis of monomers and in the preparation of polymers thanks to their low cost and low toxicity and to the possibility of their large-scale production [47].

There are four main types of epoxidation reactions to functionalize the double bonds on triglycerides. (i) The conventional process is the classic Prileschajew epoxidation reaction [48], involving the use of hydrogen peroxide, peracids or peroxyacids. The production processes for EVOs preparation mostly require performic and peracetic acids, which are generated in situ in the aqueous phase by reaction of the corresponding acid (formic or acetic) with hydrogen peroxide in the presence of a strong inorganic acid, such as sulfuric acid. The peracids obtained are then transferred into the organic phase, where the epoxidation reaction takes place leading to the formation of the epoxidized oil and to the release of formic acid [49]. The use of concentrated strong acids and the secondary reactions, which occur due to the presence of peracetic acid, released as a by-product, reduce the selectivity of the epoxidation reaction. (ii) Reactions with acidic ion exchange resins (AIERs) were proposed by Gurbanov and Mamedow [50], in which resins that exchange chlorinated cations as catalysts for the synthesis of peracids were used to overcome the problems caused by the use of concentrated sulfuric acid in traditional processes [50]. This type of catalyst allows the minimization of undesired secondary reactions and, being a solid phase catalyst, it can be easily recovered from the reaction mixture to be reused [46]. (iii) Epoxidation by chemo-enzymatic catalysis without acid catalysts represents a more sustainable reaction in the context of protecting the environment. Klaas and colleagues worked on the epoxidation of vegetable oils by exploiting a perhydrolysis process in the presence of H_2_O_2_ catalyzed by the catalytic system Novozym^®^ 435 (Novo Nordisk Biotechnologie, Mainz, Germany) composed of *Candida antarctica* lipase immobilized on polyacrylic Lewatit spheres [51]. The process is extremely selective and allows epoxidized triglycerides to be obtained without by-products with a high reaction yield. The catalytic system can be recovered and reused several times before losing activity [46]. For these types of processes, it is important to have controlled and rather mild reaction conditions to preserve the enzyme integrity and proper function. (iv) Metal-catalyzed epoxidation, where the use of metal catalysts, such as molybdenum, titanium, tungsten or rhenium, offers the advantage of accurately controlling the degree of epoxidation and enables the improvement of conversion efficiency. Gerbase and colleagues described the epoxidation of soybean oil using the biphasic MTO-H_2_O_2_/CH_2_Cl_2_ catalytic system based on methyltrioxorenium (MTO) and hydrogen peroxide in dichloromethane, which allowed to obtain 100% conversions to be obtained with good selectivity (95%) and a good degree of epoxidation (95%) [52].

EVOs are already commonly used as plasticizers for PVC (poly(vinyl chloride)), instead of phthalates, to improve its stability and flexibility [47,53]. They are also used as stabilizers, lubricants and starting materials for the production of polyols, and in the synthesis of polyurethane (PU) foams [54]. Furthermore, epoxidized soybean oil has also been used as a non-toxic and biocompatible plasticizer for PHB (poly(3-hydroxybutyrate)) and PLA [47]. In recent years, vegetable oils have been proposed for the production of polymeric materials, mainly soybean oil and linseed oil, being cheap and readily available oils. The main components of these two oils are different, being linoleic acid and linolenic acid in soybean and linseed oil, respectively. Soybean oil has the greatest conversion percentage into epoxide, with the lowest activation energy in the reaction with peracetic acid. In general, commercially available epoxidized soybean oil (ESO) shows an average of 4.1–4.6 epoxy rings per triglyceride [53]. Epoxidized linseed oil (ELO) is also commercially available and has an average of 5.5 epoxy rings per triglyceride [55], more than those present in soybean oil, as it is richer in polyunsaturated fatty acids. The presence of epoxy groups and their highly branched structure make EVOs suitable for cationic polymerization processes, which, in the presence of suitable photoinitiators, can be photochemically activated and exploited for 3D printing technology.

### 2.2. Synthesis Processes of Polymeric Biomaterials Based on Vegetable Oils

Polymers are obtained through various chemical reactions. Depending on the reaction mechanism, two classes of processes can be distinguished: chain polymerizations and step polymerizations. In the step polymerization, two molecules combine to produce a third molecule with a higher molecular weight through the elimination of another molecule with a lower molecular weight. These reactions take place between monomers that have two or more reactive sites, for example, in the condensation reactions [56]. In the chain polymerization, the reaction is instead triggered by the formation of a reactive species, which can be a radical, a carbanion or a carbocation, therefore leading to a radical, anionic or cationic reaction type, respectively. The polymeric chain elongation takes place by the addition of successive monomers to the reaction center which is located at the end of the chain.

The most frequent reactions for the synthesis of biopolymers based on vegetable oils are radical polymerization, condensation polymerization and cationic polymerization.

Radical polymerization reactions occur through three steps, i.e., activation, propagation and termination. However, there are also other radical steps, such as radical transfer, that participate in the radical termination step, called termination by disproportionation [57]. In the activation step, a free radical is generated by the homolytic cleavage of a bond on the initiator and subsequently the radical reacts with the monomers in the propagation step where chain elongation occurs. The reaction ends with the termination step through the coupling of two radicals. In the case of vegetable oils, radical reactions can also occur on the C=C double bonds present on the triglycerides of non-functionalized oils, but due to the absence of conjugation, the reaction does not occur easily. This leads to the formation of liquid and viscous polymers with low molecular weight which have limited use in the industrial field [30,58]. Therefore, it is better to choose functionalized vegetable oils and in particular acrylated oils (AEVO, acrylated epoxidized vegetable oil), which are obtained from the reaction of EVO with acrylic acid. Acrylate epoxidized soybean oil (AESO), produced by UCB Chemicals Company with the commercial name “Ebecryl 860” [58], is one of the most commonly used acrylate oils with various applications in the fields of polymers and composite materials. Polymerization reactions that occur by condensation take place on functionalized vegetable oils derivatives and can lead to the formation of polyesters, polyamides and PUs [58]. Polyesters derived from vegetable oils can be obtained by polycondensation of a dicarboxylic acid with a diol, by polycondensation of hydroxy acids, by opening of lactones or by biocatalysis using the enzymatic catalyst based on lipase from *Candida antarctica*, Novozym^®^ 435. These polymers may have applications in industrial or biomedical fields, since they are similar to PCL [58]. Polyamides derived from vegetable oils have been employed in the paint and ink industry as toners, with performances comparable to those of commercial toners [59].

Finally, there are the cationic polymerization reactions in which vegetable oils, and in particular, their epoxidized derivatives are very good reactants due to their branched structure. In fact, since each fatty acid of the triglyceride can participate in the reaction, an extensive cross-linked network is formed in the process. Furthermore, the secondary reactions that usually occur and which cause problems in many polymerization processes do not inhibit this cross-linking process, but rather allow a three-dimensional polymeric network to be obtained [58]. In the cationic polymerization reactions, the reactive species is a positively charged ion, generally a carbocation, which is generated following the heterolytic cleavage of a bond on the initiator. The elongation mechanism of the polymer involves an electrophilic attack by the carbocation, first, and then by the macro-cation on the monomer, which must have a heteroatom bonded to the attacked electron-poor carbon atom [60].

### 2.3. Cationic Photopolymerization

Depending on the type of initiator, polymerization reactions can be activated thermally or photochemically, and the materials produced by the latter can be used for 3D printing of biopolymers through hardening by UV-curing processes. The UV-curing refers to processes involving the curing of monomers or oligomers from the liquid to the solid state by the formation of cross-linked chemical bonds or by polymerization reactions using exposure to UV radiation [61]. Photopolymerization is a light-induced polymerization technique that is gaining increasing attention, as it shows many advantages compared with the traditional thermal polymerization methods, including the rapid formation of polymeric networks at room temperature, no release of volatile organic compounds (VOCs) and the notable mechanical properties of the obtained polymers [62]. However, in photopolymerization-based 3D printing the heterogeneity of the 3D printed bodies, due to either the defects printed layers or inhomogeneous conversion of the monomer throughout the printing, represents the main disadvantage leading to their poor mechanical properties [63]. To overcome these drawbacks, a fine evaluation of the curing time, print orientation, sample thickness and addition of fillers are required in combination with post-curing optimization strategies focused on the network density [63,64]. One of the main photopolymerization processes is the cationic photopolymerization, which shows interesting characteristics, being insensitive to the presence of oxygen, as opposed to the radical photopolymerization, which is instead inhibited by oxygen, and allowing fine control of the cured product. Moreover, the more commonly used monomers have both low toxicity and low irritant action [62]. However, probably due to the low propagation rate constant, the rate of cationic photopolymerization is one order of magnitude lower than that of radical photopolymerization of diacrylate monomers [65]. In the cationic photopolymerization reactions, as in all light-activated polymerization reactions, the following components are required:Monomers or oligomers, the polymer precursors, which in the cationic photopolymerization are monomers that cannot be polymerized by radical mechanisms, such as epoxides, vinyl ethers and many heterocyclic compounds [66]. The monomer can contain one or more reactive groups, and this allows either linear polymers or a three-dimensional network to be obtained. Some of the most commonly used monomers in cationic polymerizations are summarized in Table 1.Photoinitiator (PhI), which is a molecule able to form reactive species (radical or ionic), which initiate the polymerization reaction. These molecules are photosensitive molecules that absorb the light radiation and pass into an excited electronic state in which the reactive species is generated by homolytic or heterolytic cleavage of a bond. Its presence is essential when the monomers used are unable to absorb the radiation and to initiate the polymerization reaction autonomously. To properly choose the photoinitiator, it is important to consider that these molecules have a very selective light absorption and it is, therefore, necessary to use a molecule which shows high absorption capabilities in the emission range of the lamp chosen as the light source. In photochemical reactions, quantum yield is defined as the number of molecules reacted per quantum of the absorbed light. The quantum yield of the initiating species (Φi) correlates with photophysical deactivation of the excited states of PhI mainly by fluorescence and phosphorescence. Low quantum yields of fluorescence or phosphorescence of PhI lead to higher Φi, which is found to be proportional to the initiation rate of cationic photopolymerization [62]. In the cationic photopolymerization mechanism the photolysis step is dependent on the quantum yield of the PhI as well as on the intensity and wavelength of the light used [67].

Furthermore, a good PhI must have a rather short half-life to avoid quenching phenomena and to generate the reactive species with a high quantum yield [61]. Cationic photoinitiators (CPhIs) are used as molecules capable of producing cationic initiating species, i.e., positively charged ions, which initiate the polymerization reaction of the monomer. Among the most commonly used CPhIs are onium salts (On^+^X^−^), which are molecules composed of an organic cation (On^+^) with a positive charge on a heteroatom and a counter ion (X^−^). Among those there are the aryldiazonium, alkyl-arylsulfonium, phosphonium, ammonium, diaryliodonium and triarylsulfonium salts (see Figure 3). Onium salts are essentially insoluble in water, but soluble in a large variety of polar solvents and monomers suitable for cationic polymerization, and show very intense absorptions in the UV region between 220 and 250 nm. The cationic moiety of these molecules acts as a chromophore and is responsible for the characteristics related to the light absorption, such as λ_max_ and molar extinction coefficient, photochemistry, quantum yield and the thermal stability of the salt. The anion instead determines the chemical characteristics, such as nucleophilicity. It also influences the strength of the acid formed during photolysis and its efficiency of initiation and determines the propagation speed. Among the most commonly used anions are the triflate ions (CF_3_SO_3_^−^), hexafluorophosphate (PF_6_^−^), hexafluoroarsenate (AsF_6_^−^), hexafluoroantimonate (SbF_6_^−^) and hexafluoroborate (BF_6_^−^) [62,81]. The most frequently used photoinitiators that belong to this category and employed in cationic polymerization reactions are the diaryliodonium and triarylsulfonium salts, discovered for the first time as cationic photoinitiators by Crivello, in combination with poorly nucleophilic anions, such as hexafluorophosphate or hexafluoroantimonate [66,81].

The cationic photopolymerization mechanism, shown in Figure 4, is characterized by three distinct phases: the initial phase of the polymerization process, in which there is the photolysis of the onium salt and the generation of Brønsted acid, which is the initiator species, the phases of propagation, with the polymer formation, and of termination [67]. The first step involves the onium salt, which, following irradiation with the appropriate λ, passes into an electronically excited state of singlet 1[On^+^X^−^] or triplet 3[On^+^X^−^]. Then, this species undergoes the rapid cleavage of the weak bond between the carbon atom and the heteroatom (e.g., C–I, C–S, C–P), which can be homolytic, leading to the generation of a radical cation (R^٠+^), or heterolytic, generating a cation (R^+^). In the next propagation phase, the cation and/or the cation radical gains a proton from a hydrogen donor, which can be the salt itself, the monomer or another component of the reaction mixture, and a protonic acid, or Brønsted acid (H^+^X^−^), which is a highly efficient initiating species, is released. The actual beginning of the cationic polymerization reaction occurs when the Brønsted acid protonates the oxygen present on the epoxide of the monomer. The reactive species generated interacts with the other epoxides present in the reaction mixture and the opening of the epoxy ring occurs leading to the formation of the first C–C bond. In the propagation steps of the reaction, the attack by the positively charged species on the heteroatom induces the opening of the epoxide and the formation of C–C bonds to repeat, resulting in the formation of the polymer. The propagation phase could proceed by either the ACE (activated chain end) or AM (activated monomer) mechanism [82]. In the ACE mechanism, the active species, located to the end of the growing macromolecule, undergoes the nucleophilic attack by the heteroatom of the epoxy groups for chain propagation. Instead, the AM mechanism takes place when the cationic polymerization of epoxides occurs in the presence of alcohols. This mechanism involves the nucleophilic attack of chain end (OH- group) to the monomer with positive charge. The degree of polymerization (DP) is dependent of the instantaneous concentration of monomer in the system. Therefore, the AM mechanism leads to the formation of the nucleophilic end-group to make this mechanism competitive with respect to that of ACE [82].

The deprotonation of the protonated macro-monomer by the epoxide leads to the termination of chain growth and of polymer formation. The termination step is essentially due to the formation of reactive species with hydroxy end-groups that do not interact with each other. The chemical aspects of UV-induced cationic photopolymerization of epoxy monomers that employs iodonium salt PhIs and thermal radical initiators have been described in a comprehensive review by Sangermano et al. [83].

However, the use of PhIs of this type is quite restricted due to the poor overlap of their absorption spectrum with the emission range of traditional mercury lamps and especially of the most commonly used light sources for 3D printing, such as LEDs or lasers. For exploiting cationic photopolymerization in 3D printing, UV-based systems show some issues related to the low penetration depth, and therefore, to the very limited thickness of the cross-linked layer, resulting in low printing speeds and in a very slow process [84]. Furthermore, the long exposure to UV radiations can induce DNA damage and genetic alterations of the cells [29], making this type of source unusable in 3D-bioprinting. Therefore, CPhI systems with an absorption range extended to longer wavelengths, i.e., with λ_max_ > 350 nm, have recently been designed and developed. To this aim, various strategies have been adopted, such as the structural modifications of conventional onium salts, by introducing electron donors or highly conjugated functional groups, or onium salts photosensitization using photosensitizers (PS). Photosensitizers are molecules that absorb light at a certain wavelength and are able to transfer the absorbed energy to the PhI which can reach the excited state and cause photolysis, leading to the formation of the reactive species. Generally, photosensitization takes place by electron transfer between the photosensitizer, which is excited following the radiation absorption and passes to a singlet (^1^PS) or triplet (^3^PS) state, and the onium salt in the ground state. Following the transfer, the free radical on the onium salt (On^٠^) and the radical cation on the photosensitizer (PS^٠+^) are formed and the PS^٠+^ initiates the cationic polymerization reaction by withdrawing a proton from a hydrogen donor to form the Brønsted acid. Highly conjugated aromatic hydrocarbons (e.g., anthracene and perylene) or heterocyclic compounds (e.g., phenothiazine) have low Eox1/2 and high excitation energy values, resulting in negative ΔG_et_ (ΔG of electron transfer) values. Therefore, these compounds facilitate photosensitized cationic polymerization when combined with the conventional onium salts. Highly conjugated molecules, thanks to their capability of absorbing light above 400 nm have been shown to be good photoinitiators for the promotion of long wavelength cationic polymerization [62]. Among the best-known photosensitizers, there are many polycyclic hydrocarbons and heterocyclic compounds (Figure 5), such as anthracene, perylene and pyrene. There are also various dyes, both synthetic, such as boron-dipyrromethenes (BODIPY), which can be functionalized to have tunable absorption, and natural, such as coumarin and its derivatives or curcumin. The latter represents a valid alternative to the typically used organic photosensitizers, due to its lower cost, easy availability, low or absent toxicity and greater environmental sustainability [85].

## 3. Functionalization and Biomedical Applications of Vegetable Oil-Based Biomaterials

### 3.1. Functionalization of Vegetable Oil-Based Biomaterials

Generally, biopolymers derived from vegetable oils have limited applications, since they have poor mechanical properties and are often fragile, do not have adequate stiffness and strength for structural applications, have high gas and vapor permeability and have a low deflection temperature (HDT, heat distortion temperature) [86,87]. To counteract these drawbacks, functionalization represents a very effective strategy to improve or modulate the mechanical or thermal properties of these polymers and make them more similar to the standards of petroleum-derived plastic materials. It is possible to directly functionalize the monomer before the polymerization reaction by its copolymerization with other monomers or, alternatively, by adding nanometric fillers or other molecules to the liquid formulation to obtain composite biomaterials.

Traditional fillers have been used such as carbon-based nanomaterials and in particular nanotubes, fullerenes, graphene and oxidized graphene (graphene oxide), in order to improve the mechanical and thermal properties of the oil-based materials, which is possible thanks to the excellent mechanical, thermal and electric conductivity properties of these additives [88,89,90,91]. Nanostructured silicon-based fillers can also be used, as proposed by Tsujimoto and colleagues [87]. These researchers have developed a preparation method for a composite based on epoxidized vegetable oils and 3-glycidoxypropyl trimethyl-siloxane (GPTMS). The nanocomposite was obtained by acid-catalyzed thermal polymerization at 140 °C in the presence of a benzyl sulfonium hexafluoroantimonate derivative, which is a thermally latent catalyst. Latent catalysts are inert molecules under normal conditions, i.e., at room temperature, but they show activity by certain external stimuli, such as heating [92,93]. 

The nanocomposite obtained by Tsujimoto et al. allowed the production of thin films, which showed good flexibility, excellent coating properties, good biodegradability and mechanical properties of resistance and hardness improved by the incorporation of the silicon network that is generated within the organic polymer matrix [87]. Another example of a composite material with very interesting potential applications is that proposed by Miao and colleagues, who used the polymer formed from epoxidized soybean oil (PESO, poly epoxidized soybean oil) as a paper coating with water-repellent properties. During the polymerization of soybean oil, the triglyceride and fatty acid structure is preserved and this confers a high hydrophobicity to the polymer [94]. In this work, the oil was polymerized in situ by a cationic mechanism (see Figure 6), in the presence of boron trifluoride diethyl etherate (BF_3_OEt_2_) on the surface of the paper to ensure that the polymer penetrates the cellulose fibers. In this way, a transparent coating is obtained which gives water-repellent properties to the paper and offers interesting application possibilities when used as a biomaterial for packaging [94].

In epoxidized vegetable oils, the functionalization of the monomer exploits the epoxy groups present on the triglycerides and takes place through ring-opening reactions caused by the nucleophilic attack of alcohols, amines or thiols. An example of this approach is described by Albarrán-Preza and colleagues, who developed a protocol for the synthesis of epoxy resins starting from epoxidized linseed oil (ELO) and xylitol. Xylitol is a sugar with five carbon atoms (Figure 7), which is widely used in the synthesis of polymers to improve their hydrophilicity, biocompatibility and biodegradability properties. Through thermal polymerization at 180 °C, a cross-linked biopolymer of completely natural origin is obtained with possible applications in the PUs industry [95]. 

Many polymers show the “shape memory” property that enable them to remember a shape and to return to its original shape after severe deformation in response to certain external stimuli, such as temperature, pH, humidity, chemicals, light or electricity. Among them, thermoresponsive shape memory materials are the most interesting thanks to their low cost and high shape recovery ability at relative low temperatures. This property depends on their elastic modulus-temperature behavior; in particular, they can be easily deformed at temperatures above their glass transition temperature (T_g_), where they achieve a rubbery elastic state. The shape is then fixed by cooling the material below its T_g_ and the material can easily return to its original shape by reheating it to a temperature higher than the T_g_ [96,97]. Extensive knowledge about the structures, functionality, modeling, synthesis methods and applications of shape-memory polymers (SMPs) has also been achieved in the biomedical field [98,99]. Moreover, the multi-shape memory effect (multi-SME), which is the property of polymers to memorize more than two temporary shapes [100], can be obtained by amorphous SMPs with hard and soft segments, significantly improving the SMPs potential and practical applications [101]. In this context, a thermodynamic model to explain the working mechanism and thermomechanical behavior of SMPs, with tunable multi-SME, has been developed by Lu et al. [101].

Copolymers of epoxidized soybean oil (ESO) and poly-L-lactic acid (PLLA) have also been synthesized by acid-catalyzed thermal polymerization and proposed as SMPs [96]. The thermoresponsive copolymer synthesized in this work shows good mechanical properties (elastic modulus and tensile strength) thanks to the presence of PLLA and exhibits excellent shape-memory properties [96]. Moreover, the functionalization of vegetable oil-based biopolymers allows the production of biomaterials suitable not only for structural applications, but also for the fabrication of scaffolds for stem cells and tissue growth.

### 3.2. Vegetable Oil-Based Biomaterials as Scaffolds for Stem Cell Growth 

A key need for effective tissue engineering and regenerative medicine is to recreate the cellular microenvironment that serves as a mechanical and biological support for cell growth and differentiation, resembling the native extracellular matrix (ECM) [102]. Three-dimensional scaffolds (3D scaffolds) have the task of mimicking some critical aspects of the ECM, such as porosity and the release of biochemical cues, and also its mechanical properties, in order to influence the behavior of the cells (adhesion, differentiation and proliferation) [103]. Plant-based biomaterials [104,105,106], and more in particular, vegetable oil-based biopolymers, are currently emerging as sustainable and suitable biomaterials for the fabrication of 3D scaffolds for implantation. Several approaches to the 3D scaffolds fabrication using vegetable oil-based biomaterials have required a functionalization of the EVO-based material to better regulate its mechanical properties, as well as its capabilities to promote cell adhesion and to be resorbed and replaced by the tissue (Figure 8).

An interesting approach to modifying and controlling the polymer mechanical and chemical-physical properties is the copolymerization of two or more monomers. Kolanthai and colleagues [107], for example, produced a co-polyester based on epoxidized soybean oil (ESO) and sebacic acid by thermal polymerization at 120 °C using citric acid as a cross-linker and D-Mannitol to improve the mechanical properties of the material. The copolymer obtained is of completely natural origin, biodegradable, biocompatible, cyto-compatible and able to support the adhesion, proliferation and osteogenic differentiation of human mesenchymal stem cells (hMSCs). Calcium phosphate deposits were formed at day seven, indicating osteogenic differentiation, without any addition of osteo-inductive factors in the medium.

These results have demonstrated a possible application of soybean oil in bone tissue engineering. A great advantage offered by this material is that it is composed of non-toxic monomers, which are easily eliminated from the body following the degradation of the scaffold [107]. The biocompatibility of oil-based scaffolds has been also tested in vivo by Sittinger et al. through implanting soybean scaffolds in mice [108].

A 3D laser printing technique has been used to produce 3D porous scaffolds made from epoxidized acrylated soybean oil, which were able to support the growth of human bone marrow stem cells (BM-hMSCs). Moreover, the scaffolds were characterized by a shape memory effect, leading to a 4D functionality [109]. Interestingly, the soybean oil scaffolds showed cell compatibility that was similar to that of PLLA and polycaprolactone (PCL), and significantly higher than that of polyethylene glycol diacrylate (PEGDA) [109]. 

Composite vegetable oil-based materials have been also produced. Composite materials are heterogeneous materials consisting of two or more phases with different physical properties, which combine to give a material characterized by improved mechanical and chemical-physical properties relative to those of the individual constituents. In composite biomaterials based on vegetable oils, the oils are the most abundant components and represent the matrix, whereas the fillers, or reinforcements, are nanomaterials added in small amounts to improve the mechanical properties. Therefore, proteins have been used for improving the cell-biomaterial interaction and the mechanical properties. In particular, keratin fibers are used, which are recoverable from waste feathers from the avian industries. Keratin fibers have been proposed by Hong and colleagues to functionalize materials based on acrylate epoxidized soybean oil (AESO) to improve their elastic modulus, T_g_ and general mechanical properties, obtaining a material with characteristics comparable to those of the composites most commonly used in industrial or structural applications [110].

Another example of an AESO-based composite material is described by Mondal and colleagues [111], who used nano-rods of hydroxyapatite (nHA) as a filler to make a nanocomposite ink. The composite was prepared by mixing epoxidized acrylate soybean oil, nHA, and 2-hydroxyethyl acrylate (HEA) or polyethylene glycol diacrylate (PEGDA) and was 3D printed by direct ink writing (DIW) to obtain a controlled-porosity scaffold for bone tissue engineering. The notable mechanical properties together with the addition of hydroxyapatite, widely used in the regeneration and repair of bone tissue as it mimics its inorganic mineral component, allowed the production of a scaffold on which BM-hMSCs were able to adhere, proliferate and differentiate [111]. In particular, the presence of HEA in the biomaterial significantly increased the rheological properties, extensibility and printability of the nano-composite inks, resulting in homogenous and high-fidelity scaffold structures. However, although the addition of PEGDA led to improved dispersion, the apparent ultimate compressive strength (UCS), toughness values and elastic modulus were significantly decreased by the addition of either PEGDA or HEA. BM-MSCs showed excellent viability and proliferation on all three types of 3D printed scaffolds. However, the addition of PEGDA in the AESO-based biomaterial also downregulated the osteogenic differentiation of BM-MSCs compared to AESO alone and with HEA. On the contrary, the presence of HEA led to a significant upregulation of osteogenic differentiation of BM-MSCs [111].

Recently, the unique properties of both carbohydrates and soybean oil have also been combined to create low-cost epoxy bio-resins of interest to the biomedical industry [112,113]. β-cyclodextrins (Cd) were combined with ESO (CdHS-ESO 50/50), producing a soft and flexible biomaterial able to support cell adhesion and growth (see Figure 9), with potential applications in implants and wound dressing [108] In this study, either trehalose- (Tr) or β-cyclodextrin- (Cd) based materials were produced by reaction with succinic anhydride (S) and heptanoyl chloride (H) in dimethyl formamide and pyridine at 90 °C and then cross-linked with ESO (Figure 9). Both TrHS-ESO and CdHS-ESO were characterized, demonstrating that they were soft and flexible materials, but they were quickly degraded in basic aqueous conditions. Both the epoxide materials were used as cellular scaffolds, but only the CdHS-ESO material was able to improve the adhesion and proliferation of neonatal human dermal fibroblasts (nHDFs) [112].

Composite sunflower oil-based biomaterials have been also prepared via thiol-ene Michael addition [114] and reinforced by the addition of different compositions of cellulose nanocrystals with potential applications as biosensors and scaffolds for cell proliferation. Bio-nanocomposites made of castor oil-based films and chitosan-modified ZnO nanoparticles with increased water absorption, thermal stability and barrier, mechanical, viscoelastic, wound-healing and antibacterial properties have been fabricated as very promising wound-healing materials [115].

Castor oil (CO), which is a vegetable-oil obtained from the *Ricinus communis* and characterized by the presence of triglycerides with hydroxylated groups, has also been used as a chain extender for the production of SMPs [116]. CO, used as a chain extender in PUs synthesis by polytetramethylene glycol (PTMG250), was able to improve the shape memory capacity of the PU, increasing its T*_g_* and biocompatibility. The epithelial cells (HEK293 cell line) seeded on this SMP, after 72 h of cell growth, showed an adherent spindle shape and normal morphology by microscopic analyses, demonstrating that this biomaterial was suitable for applications in biomedicine [116]. In conclusion, the EVO/AEVO-based scaffolds with appropriate functionalization to enhance their natural properties represent a novel, low-cost alternative to current medical applications, especially in bone repair. Due to their shape memory property, EVO-based SMPs can have applications in biomedical field not only as scaffolds for tissue engineering, but also as drug delivery systems and medical devices for minimally invasive surgery, i.e., removable intravascular and ureter stents, degradable sutures and orthodontic tools.

## Figures and Tables

**Figure 1 ijms-24-02153-f001:**
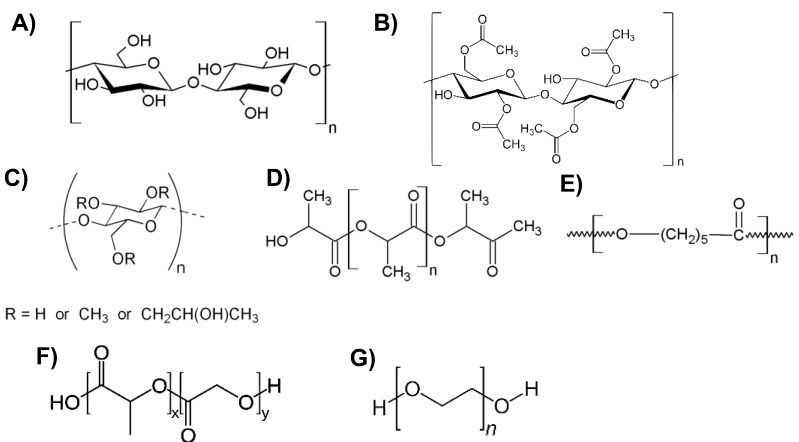
Chemical structures of cellulose (**A**), cellulose acetate (**B**), hydroxypropyl methylcellulose (**C**), polylactic acid (**D**), polycaprolactone (**E**), poly(lactic-co-glycolic) acid (**F**), polyethylene glycol (PEG) (**G**).

**Figure 2 ijms-24-02153-f002:**
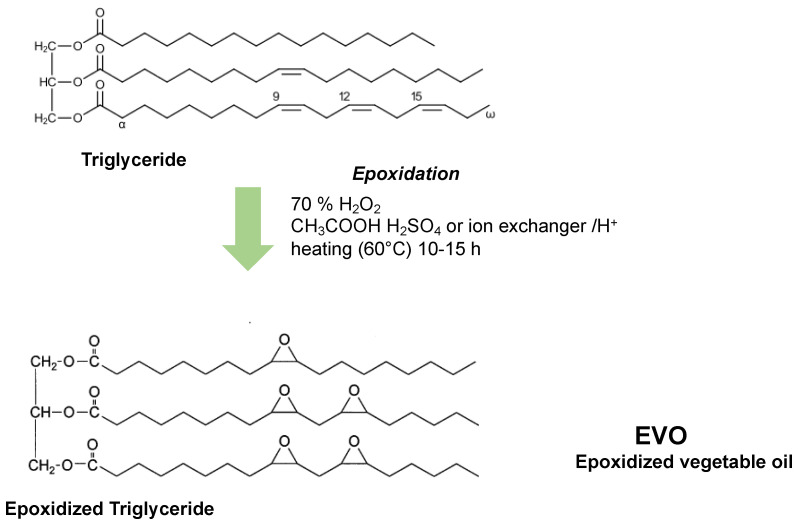
Chemical structure representation of triglycerides and epoxidation by Prileschajew’s reaction [48].

**Figure 3 ijms-24-02153-f003:**
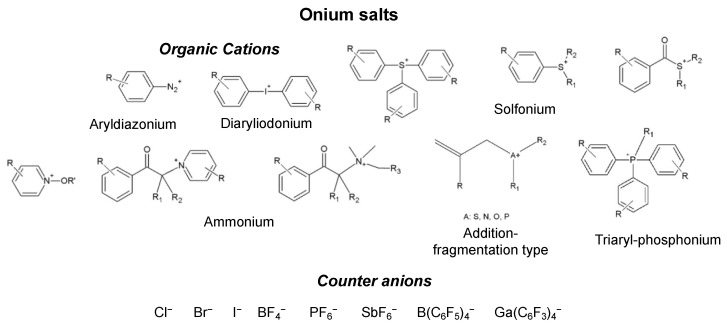
Chemical structures of the most commonly used onium salts.

**Figure 4 ijms-24-02153-f004:**
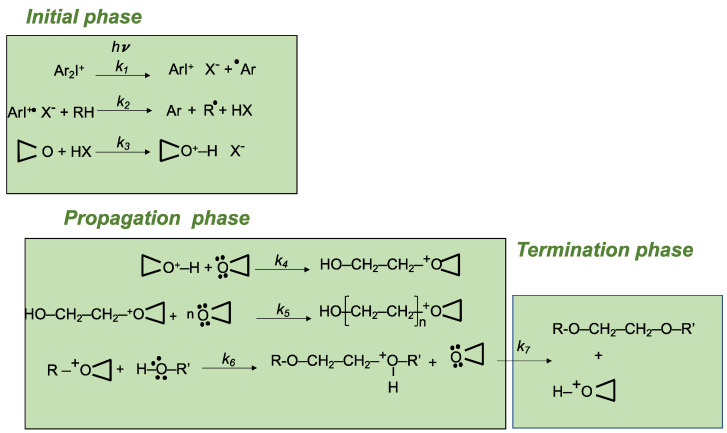
Schematic representation of the cationic polymerization process involving a diaryliodonium salt (Ar-I-Ar) as CPhI (inspired from Crivello et al.) [67] (k is the kinetic rate constant).

**Figure 5 ijms-24-02153-f005:**
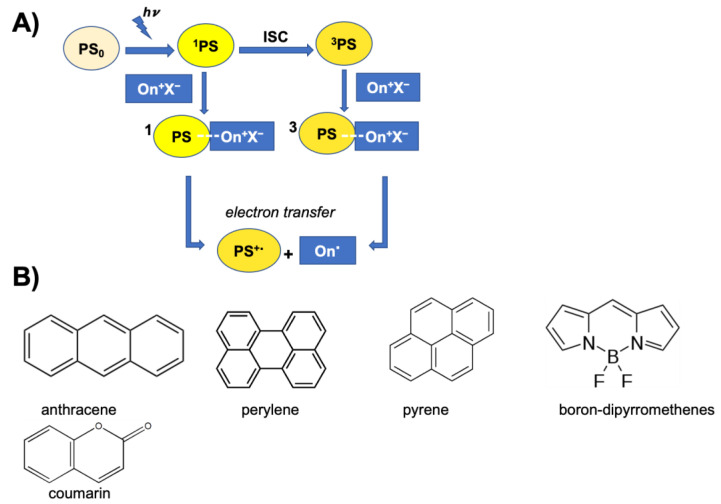
(**A**) Schematic representation of the onium salt photosensitization process by electron transfer. (**B**) Polycyclic aromatic compounds most commonly used as photosensitizers: anthracene, perylene, pyrene, boron-dipyrromethenes, coumarin. ^1^PS photosensitizer in a singlet or ^3^PS triplet state, On^٠^ radical onium salt.

**Figure 6 ijms-24-02153-f006:**
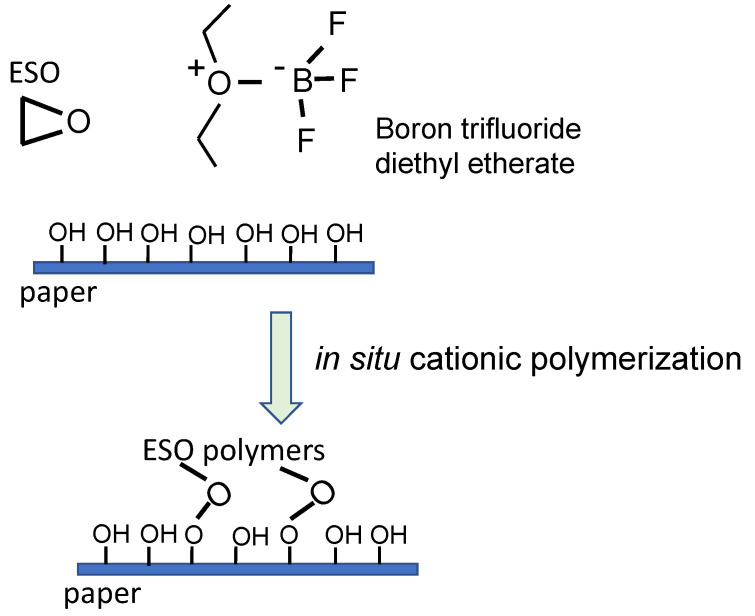
Schematic illustration of the in situ cationic polymerization for the preparation of the ESO-based paper composites (inspired from Miao, S et al.) [94].

**Figure 7 ijms-24-02153-f007:**
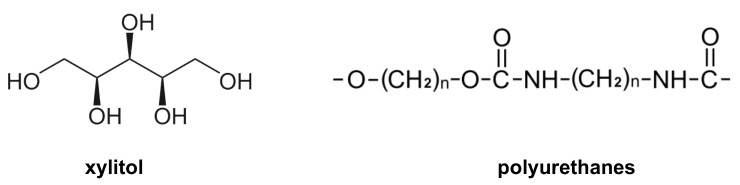
Chemical structures of the xylitol and polyurethanes (PUs).

**Figure 8 ijms-24-02153-f008:**
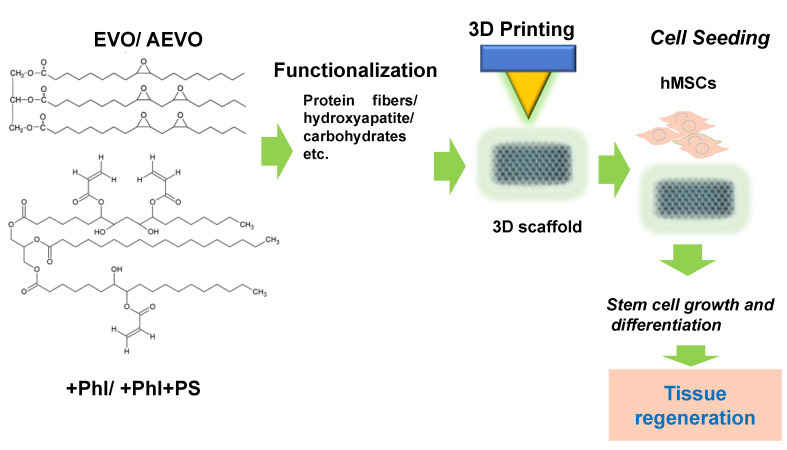
Schematic representation of the vegetable oil-based scaffolds for stem cell growth.

**Figure 9 ijms-24-02153-f009:**
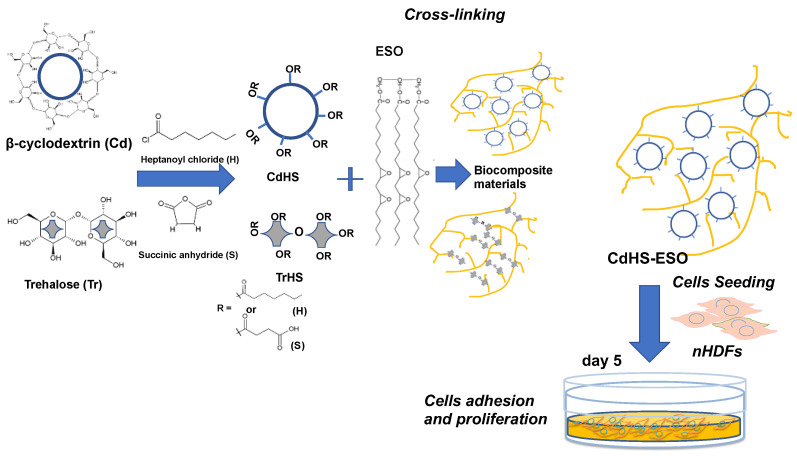
Schematic representation of the CdHS-ESO- and TrHS-ESO-based composite materials production and the cell scaffolding property of CdHS-ESO. H, heptanoyl chloride; S, succinic anhydride; CdHS, β-cyclodextrin with ∼6 H units and ∼8 S units; TrHS, trehalose with ~4 H units and ~3 S units; nHDFs, neonatal human dermal fibroblasts (inspired from Zhang, Q. et al.) [112].

**Table 1 ijms-24-02153-t001:** Monomers used in cationic polymerizations.

Monofunctional Monomers	Difunctional Monomers	*Ref*.
Epoxide 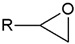	EC 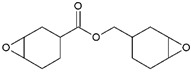	[68,69,70]
Vinyl ether 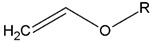	di-epoxide 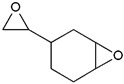	[68,71]
CHO 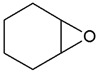	SIB 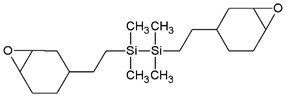	[72,73]
NVK 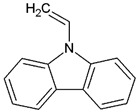	Epalloy 5000 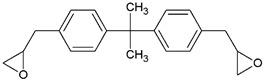	[74,75]
THF 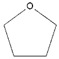	DVE-3 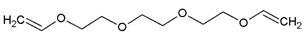	[76,77]
Caprolactone 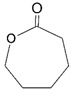		[78]
Styrene 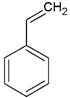		[79]
Propylene carbonate 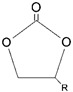		[80]

## Data Availability

Not applicable.

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
