# Peer review of "Sustainable Vegetable Oil-Based Biomaterials: Synthesis and Biomedical Applications"

_ijms, 2023, doi:10.3390/ijms24032153_

Round 1

Reviewer 1 Report

l. 73-74 whether these compounds can really be considered Krebs cycle intermediates?

Figure 4 please explain k symbol

l.344 please correct "mechanica,l" 

Author Response

POINT-TO-POINT response to Reviewer’s Comments

First of all, we would like to thank the reviewers for their constructive and positive comments. We hope that our efforts to address the reviewers’ concerns have succeeded in allowing us to prepare a significantly better version of our manuscript. We have devoted a significant time to follow their useful suggestions and address all the comments point-by-point, as shown below.

We hope that the new amended manuscript will meet the requirement of the journal and the interest of the scientific community.

Referee 1

Comments and Suggestions for Authors

  1. 73-74 whether these compounds can really be considered Krebs cycle intermediates?

Figure 4 please explain k symbol

l.344 please correct "mechanica,l" 

Answer: We thank to the referee for these comments. We have changed the manuscript accordingly.

Reviewer 2 Report

1) The review is poor in the number of references and comments for natural polymers used in 3D printing technology and scaffolds.The authors would add relevant information to this review by reading the following articles (not included in their bibliographic reference)
    - Vegetable-oil-based polymers as future polymeric biomaterials (10.1016/j.actbio.2013.08.040)
     - Polymers and Composites Derived from Castor Oil as Sustainable Materials and Degradable Biomaterials: Current Status and Emerging Trends (10.1021/acs.biomac.0c01291)
     - Vegetable Oil-Based Polymers: Properties, Processing and Applications De Niranjan Karak (https://www.elsevier.com/books/vegetable-oil-based-polymers/karak/978-0-85709-710-1)
     - A review on recent approaches to sustainable bio-based epoxy vitrimer from epoxidized vegetable oils (10.1016/j.indcrop.2022.115857
     - Sustainable Series of New Epoxidized Vegetable Oil-Based Thermosets with Chemical Recycling Properties (10.1021/acs.biomac.0c01059)
     -  A Review of Vegetable Oil-Based Polymers: Synthesis and Applications (DOI: 10.4236/OJPCHEM.2015.53004)
     - CHAPTER 1 Photo-cured Materials from Vegetable Oils (DOI: 10.1039/9781782621850-00001)
     - Sustainable Polymeric Materials Derived From Plant Oils: From Synthesis To ApplicationsSynthesis To Applications. (https://scholarcommons.sc.edu/cgi/viewcontent.cgi?article=5569&context=etd)
     - Flexible polymeric biomaterials from epoxidized soybean oil, epoxidized oleic acid, and citric acid as both a hardener and acid catalyst (10.1002/app.53011)
     - A Review of 3D Printing Technology for Medical Applications (DOI: 10.1016/j.eng.2018.07.021)

2) L37-38 The authors should mention why there are limitations regarding the modulation of the chemical-physical and mechanical properties of natural polymers.
3) What do you mean "ex-novo"? I couldn't understand the sentence L39-41.
4) L58-59 ... lactic acid monomers, which can also be extracted from renewable sources such as corn starch or from waste from the agri-food chain. Here, the authors should mention how lactic acid is extracted. There are biochemical processes to convert glucose into lactic acid whose authors do not mention, see (10.1128/AEM.01514-08)
5) L59-65. Although PLA has some interesting properties for use as scaffolds and for 3D printing, this polymer is not as good at tissue engineering as the authors report. Once PLA hydrolytically degrades in the body to form lactic acid that induces inflammatory responses
(https://doi.org/10.1186/s13036-017-0074-3)
6) L66-71. Regarding PCL, authors should read the review: Polycaprolactone as a biomaterial for bone scaffolds: literature review (10.1016/j.jobcr.2019.10.003) and make comments such as improvement of hydrophilicity, a drawback of this polymer in the production of scaffolds to promote cell growth
7) L73. PLGA hydrolysis does not produce lactate, but lactic acid. Like PLA, such hydrolysis is a major drawback in the production of scaffolds, due to the production of acidic monomers that provide a local immune response despite the removal of these monomers by the Krebs cycle.
8) L78-80. The authors do not discuss the drawbacks of PEG such as poor mechanical strength and cell attachment limit. Instead, it is often used as an additive, compound, or delivery system (10.1615/jlongtermeffmediimplants.2013006244 )
9) L99-101. The authors should mention the production of biofuels from oils by transesterification and show a reaction thereof. (10.1016/j.ceja.2022.100284)
10) L159-161. Authors should provide a reference and a scheme detailing the cationic mechanism.
11) L164-176. These types of reactions are well known to the polymer community. Please cite Flory's Book.
12) L177-181.In addition to initiation, propagation, and termination steps, there are other radical steps such as radical transfer. This type of reaction also participates in the radical termination step called termination by disproportionation. Please state them in the text.
13) L217-220. One of the main drawbacks of photopolymerization-based 3D printing systems is the brittleness of the final product due to the formation of inhomogeneous polymer networks with high crosslinking densities (https://polyspectra.com/blog/photopolymers-in- 3d- printing/).
Why don't athours discuss these drawbacks and how to overcome them?
14) L228-229. Authors should discuss which polymerization rate is higher, cationic or radical (see Ring-opening polymerization and Special Polymerization Processes P. Kubisa, in Polymer Science: A Comprehensive Reference, 2012).
15) L248. Explain what is quantum yield and initiation efficiency. These parameters have a great influence on the rate of polymerization and the molecular weight of the polymers.
16) How is the molecular weight and crosslinking of cationic polymerizations compared to radical polymerizations? This fact influences the mechanical and thermophysical properties of the materials used as biomaterial.
17) The mechanism represented in Figure 4 has low resolution. I couldn't differentiate the cation from the radical form.
18) In the text, the authors should state that there are two polymerization mechanisms: AM and ACE. Here, authors should compare PD between two mechanisms (PCR I Stanislaw Penczek, Przemyslaw Kubisa, in Comprehensive Polymer Science and Supplements, 1989; see also 10.3144/expresspolymlett.2018.78)
19) In Figure 4, authors should include the termination step. Is this a living polymerization?
20) L316. Why are polycyclic hydrocarbons and heterocyclic compounds used as photosensitizers? What is the difference of these compounds in relation to other non-polycyclic ones?
21) L344. correct mechanica,l.
22) L343. .... thanks to their excellent mechanica,l thermal and electric conductivity properties. Please explain why fullerenes, graphene and oxidized graphene (graphene oxide) provide these properties.
23) L347Explain in this manuscript what is acid catalyzed thermal polymerization and how it differs from other types of polymerization. This type of polymerization is used later in the text (section 3.1 and 3.2). The authors assume that the readers are experts in polymerization. Because this is a review, details must be given.
24) L356-357. Include the cationic mechanism of the paper [55]
25) L365. Include a figure for the structure of xylitol
26) L370. Include a figure for polyurethanes
27) L374-376. The authors should comment here on the memory-recovery mechanism.
28) L409-412. The authors at the beginning of the text (possibly in the introduction) should indicate the difference between 3D and 4D printing
29) L423-428. Acrylate epoxidized soybean oil (AESO)-based materials (ref. 63) is not used for biomedical applications, but for electronic as well as automotive and aeronautical applications. In addition, acrylates and methacrylates show cytotoxicity (10.1002/(sici)1097-4636(19971215)37:4<517::aid-jbm10>3.0.co;2-5), therefore, it is out of context of biomaterials.
However, the authors could discuss this drawback in terms of structure and use as an (AESO)-based biomaterial.
30) I think in the ref. [64] were evaluated mechanical properties and cytocompatibility of AESO with nano-Hydroxyapatite (nHA) rods and either 2-Hydroxyethyl Acrylate (HEA) or Polyethylene Glycol Diacrylate (PEGDA as function of functional groups. The authors could tackle this interesting point in the review.
31) L437-439. The authors could include the reaction used to obtain sustainable and degradable epoxy resins from Trehalose, Cyclodextrin, and Soybean Oil Yield (ref. [65]) and discuss it.

Author Response

POINT-TO-POINT response to Reviewer’s Comments

First of all, we would like to thank the reviewers for their constructive and positive comments. We hope that our efforts to address the reviewers’ concerns have succeeded in allowing us to prepare a significantly better version of our manuscript. We have devoted a significant time to follow their useful suggestions and address all the comments point-by-point, as shown below. 
We hope that the new amended manuscript will meet the requirement of the journal and the interest of the scientific community.

Referee 2

Open Review
( ) I would not like to sign my review report
(x) I would like to sign my review report
English language and style
( ) English very difficult to understand/incomprehensible
(x) Extensive editing of English language and style required
( ) Moderate English changes required
( ) English language and style are fine/minor spell check required
( ) I don't feel qualified to judge about the English language and style
Is the work a significant contribution to the field?    X
Is the work well organized and comprehensively described?    X
Is the work scientifically sound and not misleading?    X
Are there appropriate and adequate references to related and previous work?    X
Is the English used correct and readable?    XXXX

Answer: We have improved the English language and style of the manuscript using the suggested MDPI English editing service and we include the English editing certificate   

Comments and Suggestions for Authors
1)    The review is poor in the number of references and comments for natural polymers used in 3D printing technology and scaffolds. The authors would add relevant information to this review by reading the following articles (not included in their bibliographic reference)
- Vegetable-oil-based polymers as future polymeric biomaterials (10.1016/j.actbio.2013.08.040)
 - Polymers and Composites Derived from Castor Oil as Sustainable Materials and Degradable Biomaterials: Current Status and Emerging Trends (10.1021/acs.biomac.0c01291)
 - Vegetable Oil-Based Polymers: Properties, Processing and Applications De Niranjan Karak (https://www.elsevier.com/books/vegetable-oil-based-polymers/karak/978-0-85709-710-1)
- A review on recent approaches to sustainable bio-based epoxy vitrimer from epoxidized vegetable oils (10.1016/j.indcrop.2022.115857
 - Sustainable Series of New Epoxidized Vegetable Oil-Based Thermosets with Chemical Recycling Properties (10.1021/acs.biomac.0c01059)
-  A Review of Vegetable Oil-Based Polymers: Synthesis and Applications (DOI: 10.4236/OJPCHEM.2015.53004)
- CHAPTER 1 Photo-cured Materials from Vegetable Oils (DOI: 10.1039/9781782621850-00001)
- Sustainable Polymeric Materials Derived From Plant Oils: From Synthesis To ApplicationsSynthesis To Applications. (https://scholarcommons.sc.edu/cgi/viewcontent.cgi?article=5569&context=etd)
 - Flexible polymeric biomaterials from epoxidized soybean oil, epoxidized oleic acid, and citric acid as both a hardener and acid catalyst (10.1002/app.53011)
- A Review of 3D Printing Technology for Medical Applications (DOI: 10.1016/j.eng.2018.07.021)
Answer: We thank the referee for this comment and in agreement we have included the following references as suggested by the referee 1:

References 

32. Miao, S.; Wang, P.; Su, Z.; Zhang, S. Vegetable-Oil-Based Polymers as Future Polymeric Biomaterials. Acta Biomaterialia 2014, 10, 1692–1704, doi:10.1016/j.actbio.2013.08.040.
33. Adekunle, K. A Review of Vegetable Oil-Based Polymers: Synthesis and Applications. Open Journal of Polymer Chemistry 2015, 5, 34-40, doi:10.4236/ojpchem.2015.53004.
35.Chakraborty, I.; Chatterjee, K. Polymers and Composites Derived from Castor Oil as Sustainable Materials and Degradable Biomaterials: Current Status and Emerging Trends. Biomacromolecules 2020, 21, 4639–4662, doi:10.1021/acs.biomac.0c01291.
36. Karak, N. 2 - Biodegradable Polymers. In Vegetable Oil-Based Polymers; Karak, N., Ed.; Woodhead Publishing, 2012; pp. 31–53 ISBN 978-0-85709-710-1.
37. Chong, K.L.; Lai, J.C.; Rahman, R.A.; Adrus, N.; Al-Saffar, Z.H.; Hassan, A.; Lim, T.H.; Wahit, M.U. A Review on Recent Approaches to Sustainable Bio-Based Epoxy Vitrimer from Epoxidized Vegetable Oils. Industrial Crops and Products 2022, 189, 115857, doi:10.1016/j.indcrop.2022.115857.
38.Di Mauro, C.; Malburet, S.; Genua, A.; Graillot, A.; Mija, A. Sustainable Series of New Epoxidized Vegetable Oil-Based Thermosets with Chemical Recycling Properties. Biomacromolecules 2020, 21, 3923–3935, doi:10.1021/acs.biomac.0c01059.
39. Gan, Y.; Jiang, X. CHAPTER 1 Photo-Cured Materials from Vegetable Oils. In Green Materials from Plant Oils; The Royal Society of Chemistry, 2015; pp. 1–27 ISBN 978-1-84973-901-6.
40. Zhang, X. Sustainable Polymeric Materials Derived From Plant Oils: From Synthesis To Applications (Master's thesis), 2017. Retrieved from https://scholarcommons.sc.edu/etd/4540.
41. Hood, C.; Ghazani, S.M.; Marangoni, A.G.; Pensini, E. Flexible Polymeric Biomaterials from Epoxidized Soybean Oil, Epoxidized Oleic Acid, and Citric Acid as Both a Hardener and Acid Catalyst. Journal of Applied Polymer Science 2022, 139, e53011, doi:10.1002/app.53011.

2)    L37-38 The authors should mention why there are limitations regarding the modulation of the chemical-physical and mechanical properties of natural polymers.
Answer: Often the stiffness of the natural biopolymers is not tunable and, in many cases, there is also a too rapid biodegradability In agreement to the comment we have added the following sentence:

Page 1 line 40
Often the stiffness of the natural biopolymers is not tunable and, in many cases, they are also rapidly biodegraded.
3)    What do you mean "ex-novo"? I couldn't understand the sentence L39-41.
Answer: Thank to the referee for this comment we have changed the “ex-novo” with “de novo” that is more used.

4)    L58-59 ... lactic acid monomers, which can also be extracted from renewable sources such as corn starch or from waste from the agri-food chain. Here, the authors should mention how lactic acid is extracted. There are biochemical processes to convert glucose into lactic acid whose authors do not mention, see (10.1128/AEM.01514-08)
Answer: Thank to the referee for this comment we and in agreement we have included the following references in the manuscript:

4) Okano Kenji; Zhang Qiao; Shinkawa Satoru; Yoshida Shogo; Tanaka Tsutomu; Fukuda Hideki; Kondo Akihiko Efficient Production of Optically Pure D-Lactic Acid from Raw Corn Starch by Using a Genetically Modified l-Lactate Dehydrogenase Gene-Deficient and α-Amylase-Secreting Lactobacillus Plantarum Strain. Applied and Environmental Microbiology 2009, 75, 462–467, doi:10.1128/AEM.01514-08.
5) Okano, K.; Uematsu, G.; Hama, S.; Tanaka, T.; Noda, H.; Kondo, A.; Honda, K. Metabolic Engineering of Lactobacillus Plantarum for Direct L-Lactic Acid Production From Raw Corn Starch. Biotechnology Journal 2018, 13, 1700517, doi:10.1002/biot.201700517.
6) Okano, K.; Zhang, Q.; Yoshida, S.; Tanaka, T.; Ogino, C.; Fukuda, H.; Kondo, A. D-Lactic Acid Production from Cellooligosaccharides and β-Glucan Using l-LDH Gene-Deficient and Endoglucanase-Secreting Lactobacillus Plantarum. Applied Microbiology and Biotechnology 2010, 85, 643–650, doi:10.1007/s00253-009-2111-8.
7) Tateno, T.; Fukuda, H.; Kondo, A. Direct Production of L-Lysine from Raw Corn Starch by Corynebacterium Glutamicum Secreting Streptococcus Bovis α-Amylase Using CspB Promoter and Signal Sequence. Applied Microbiology and Biotechnology 2007, 77, 533–541, doi:10.1007/s00253-007-1191-6.
8) Aso, Y.; Hashimoto, A.; Ohara, H. Engineering Lactococcus Lactis for D-Lactic Acid Production from Starch. Current Microbiology 2019, 76, 1186–1192, doi:10.1007/s00284-019-01742-4.
9) Zhang, Y.; Vadlani, P.V.; Kumar, A.; Hardwidge, P.R.; Govind, R.; Tanaka, T.; Kondo, A. Enhanced D-Lactic Acid Production from Renewable Resources Using Engineered Lactobacillus Plantarum. Applied Microbiology and Biotechnology 2016, 100, 279–288, doi:10.1007/s00253-015-7016-0.

5) L59-65. Although PLA has some interesting properties for use as scaffolds and for 3D printing, this polymer is not as good at tissue engineering as the authors report. Once PLA hydrolytically degrades in the body to form lactic acid that induces inflammatory responses
(https://doi.org/10.1186/s13036-017-0074-3)
Answer: Although the exacerbated inflammatory response may lead to rejection of an implant, recent studies have pointed towards a decisive role of inflammation in triggering tissue repair and regeneration. For this reason, we didn’t talk this concept in this review that is focused on oil-based biomaterials. However, in agreement with the referee we have discuss this point in the manuscript and added the following references:
1.    Journal of Biological Engineering volume 11, 31 (2017) https://doi.org/10.1186/s13036-017-0074-3
2.    Rahul MR, Amol VJ, Douglas EH. Poly (lactic acid) modifications. Prog Polym Sci. 2010;35:338–56.
3.    Catarina R. Almeida,Tiziano Serra,Marta I. Oliveira,Josep A. Planell,Melba Navarro Impact of 3-D printed PLA- and chitosan-based scaffolds on human monocyte/macrophage responses: Unraveling the effect of 3-D structures on inflammation February 2014, Volume10(Issue2)Pages, p.613To - 622 - Acta Biomaterialia

6) L66-71. Regarding PCL, authors should read the review: Polycaprolactone as a biomaterial for bone scaffolds: literature review (10.1016/j.jobcr.2019.10.003) and make comments such as improvement of hydrophilicity, a drawback of this polymer in the production of scaffolds to promote cell growth
Answer:  In agreement with the referee comment we have included the relevance of the hydrophilicity of the PCL scaffold and the suggested reference: 

10.1016/j.jobcr.2019.10.003

7) L73. PLGA hydrolysis does not produce lactate, but lactic acid. Like PLA, such hydrolysis is a major drawback in the production of scaffolds, due to the production of acidic monomers that provide a local immune response despite the removal of these monomers by the Krebs cycle.
Answer: Usually in the biological environment the acid is rapidly transformed in salt for the presence of ions so the lactic acid is present as lactate. However, we are in agreement with the referee that the monomer produced is lactic acid. 

8) L78-80. The authors do not discuss the drawbacks of PEG such as poor mechanical strength and cell attachment limit. Instead, it is often used as an additive, compound, or delivery system (10.1615/jlongtermeffmediimplants.2013006244 )
Answer: We think that main limit of the PEG is the low interaction with the cells that is bypassed by the functionalization with proteins/peptides riches in RGD sequences. Therefore, in agreement with the referee’s comment we have added the suggested reference and discuss in the manuscript the drawback of the PEG as follows:
Page 3 Line 99 
Polyethylene glycol (PEG) (Figure 1G) is a hydrophilic molecule that although shows limited interaction with cells it is characterized by hydroxylic groups that can be easily functionalized, e.g., carboxylate, thiolate, acrylate etc., in order to promote both the photopolymerization process and the binding of bioactive molecules for improving the cellular adhesion [10,11]. PEG based-biomaterials show an easily tunable stiffness, are biocompatible and not immunogenic and due to their specific properties are useful for photopolymerizable cellular scaffolds, implants and for the production of drug-releasing systems in biomedicine
Ref.  11) jlongtermeffmediimplants.2013006244

9) L99-101. The authors should mention the production of biofuels from oils by transesterification and show a reaction thereof. (10.1016/j.ceja.2022.100284)
Answer: This is a review focused on the oil-based biomaterials and their potential biomedical applications therefore we didn’t describe the relevance of the waste oils for the production of biofuel. However, in agreement with the referee’s comment we have mentioned this relevant use of the waste oils including the following sentence into the manuscript:
Page 3 Line  Therefore, waste oils derived from the agri-food industry can be used for the production of both biofuel by transesterification [10.1016/j. ceja.2022.100284] and biopolymers.
ref. 10.1016/j.ceja.2022.100284

10) L159-161. Authors should provide a reference and a scheme detailing the cationic mechanism.
Answer: The cationic mechanism was described in details in the par. 2.3 

12) L177-181.In addition to initiation, propagation, and termination steps, there are other radical steps such as radical transfer. This type of reaction also participates in the radical termination step called termination by disproportionation. Please state them in the text.
Answer: In agreement to the referee‘s comment we have changed the manuscript as follows: 
Page 5 line 199
Radical polymerization reactions occur through three steps, which are: activation, propagation and termination. However, there are also other radical steps such as radical transfer. participates in the radical termination step called termination by disproportionation [57]

14) L228-229. Authors should discuss which polymerization rate is higher, cationic or radical (see Ring-opening polymerization and Special Polymerization Processes P. Kubisa, in Polymer Science: A Comprehensive Reference, 2012).
Answer :  In agreement we have discuss this point in to the manuscript as follows: 
 Page  line 
However, probably due to the low propagation rate constant, the rate of cationic photopolymerization is by 1 order of magnitude lower than that of radical photopolymerization of diacrylate monomers [Ring-opening polymerization and Special Polymerization Processes P. Kubisa, in Polymer Science: A Comprehensive Reference, 2012].
Ref . 
Ring-opening polymerization and Special Polymerization Processes P. Kubisa, in Polymer Science: A Comprehensive Reference, 2012

17) The mechanism represented in Figure 4 has low resolution. I couldn't differentiate the cation from the radical form.
19) In Figure 4, authors should include the termination step. Is this a living polymerization?
Answer to these comments: In agreement with the referee’s comments the cationic mechanism has been described with more details in the manuscript and the figure 4 has been modified as follows :
Page 9
The cationic photopolymerization mechanism, shown in Figure 4, is characterized by three distinct phases: the initial phase of the polymerization process, in which there is the photolysis of the onium salt and the generation of Brønsted acid, which is the initiator species, and the phases of propagation, with the polymer formation, and of termination [50].

Page 9 line 6XX The termination step is essentially due to the formation of reactive species with hydroxy end-groups that do not interact with each other. The chemical aspects of UV-induced cationic photopolymerization of epoxy monomers that employ iodonium salt photoinitiators and thermal radical initiators have been currently described in a comprehensive review by Sangermano et al [Malik et al 2020].

Figure 4

Currently, the living cationic polymerization is obtainable using other kind of epoxide monomers but not using EVO.

We have also added the following references: 
Ring-opening polymerization and Special Polymerization Processes P. Kubisa, in Polymer Science: A Comprehensive Reference, 2012

Malik et al 2020

11) L164-176. These types of reactions are well known to the polymer community. Please cite Flory's Book.
Answer: In agreement to the referee’s comment, we have added the following reference:
Paul J. Flory. Principles of Polymer Chemistry Cornell University Press, 1953 - Technology & Engineering - 672 pages.

13) L217-220. One of the main drawbacks of photopolymerization-based 3D printing systems is the brittleness of the final product due to the formation of inhomogeneous polymer networks with high crosslinking densities (https://polyspectra.com/blog/photopolymers-in- 3d- printing/).Why don't athours discuss these drawbacks and how to overcome them?
Answer: In agreement to this comment, we have discussed this point in the manuscript as follow:
Page 6 line 
However, in photopolymerization-based 3D printing the heterogeneity of the 3D printed bodies, due to either the defects printed layers or inhomogeneous conversion of the monomer throughout the printing, represents the main disadvantage leading to their poor mechanical properties.[ref. Štaffová, M.; Ondreáš, F.; Svatík, J.; Zbončák, M.; Jančář, J.; Lepcio, P. 3D Printing and Post-Curing Optimization of Photopolymerized Structures: Basic Concepts and Effective Tools for Improved Thermomechanical Properties. Polymer Testing 2022, 108, 107499, doi:10.1016/j.polymertesting.2022.107499]. To solve these drawbacks, a fine evaluation of the curing time, print orientation, sample thickness and addition of fillers are required in combination with post-curing optimization strategies focused on the network density. [ref. Štaffová, M.; Ondreáš, F.; Svatík, J.; Zbončák, M.; Jančář, J.; Lepcio, P. 3D Printing and Post-Curing Optimization of Photopolymerized Structures: Basic Concepts and Effective Tools for Improved Thermomechanical Properties. Polymer Testing 2022, 108, 107499, doi:10.1016/j.polymertesting.2022.107499; LIu et al 2019]. One of the main photopolymerization processes is the cationic photopolymerization, that shows… 

15) L248. Explain what is quantum yield and initiation efficiency. These parameters have a great influence on the rate of polymerization and the molecular weight of the polymers.
Answer: In agreement with the referee’s comment, we have explained these parameters in the manuscript as follows:  
Page line 
In photochemical reactions quantum yield is defined as the number of molecules reacted per quantum of the absorbed light. The quantum yield of the initiating species (Φi) correlates with photophysical deactivation of the excited states of onium salts mainly by fluorescence and phosphorescence. Low quantum yields of fluorescence or phosphorescence of onium salts lead to higher Φi, which is found to be proportional to the initiation rate of cationic photopolymerization. [ref 34. Shi et al]
In the cationic photopolymerization mechanism the photolysis step is dependent on the quantum yield of the photoinitiator as well as on the intensity and wavelength of the light used. [ref 50. Crivello et al 2000]

16) How is the molecular weight and crosslinking of cationic polymerizations compared to radical polymerizations? This fact influences the mechanical and thermophysical properties of the materials used as biomaterial.
Answer: Obviously, both the molecular weight of the monomers and cross-linking have an influence on the mechanical and thermophysical properties of the materials. However, at this moment, there are not studies where ESO-based biomaterials, with monomers of the same molecular weight, are obtained using either the cationic and radical polymerization and are compared for their mechano-physical properties. 

18) In the text, the authors should state that there are two polymerization mechanisms: AM and ACE. Here, authors should compare PD between two mechanisms (PCR I Stanislaw Penczek, Przemyslaw Kubisa, in Comprehensive Polymer Science and Supplements, 1989; see also 10.3144/expresspolymlett.2018.78)
Answer:  We thank to the referee for this comment that give us the opportunity to explain better the mechanism. Therefore we have changed the manuscript including the following descrition: 
Page 9 line 337
The propagation phase could proceed by either ACE (activated chain end) or AM (activated monomer) mechanism [Penczek 1989 Comprehensive Polymer Science and Supplements]. In ACE mechanism the active species, located to the end of the growing macromolecule, undergoes the nucleophilic attack by the heteroatom of the epoxy groups for chain propagation. Instead, the AM mechanism takes place when the cationic polymerization of epoxides is occurs in the presence of alcohols. This mechanism involves the nucleophilic attack of chain end (OH-group) to the monomer with positive charge. The degree of polymerization (DP) is dependent of the instantaneous concentration of monomer in the system. Therefore, the AM mechanism leads to the formation of the nucleophihc end-group to make this mechanism competitive with respect to that of ACE [82].

ref . PCR I Stanislaw Penczek, Przemyslaw Kubisa, in Comprehensive Polymer Science and Supplements, 1989; see also 10.3144/expresspolymlett.2018.78

20) L316. Why are polycyclic hydrocarbons and heterocyclic compounds used as photosensitizers? What is the difference of these compounds in relation to other non-polycyclic ones?
Answer: We have described the PS, but we think that the description of the differences could be out from finality of this review. However, in agreement to the referee’s comment we have described more in details as follows: 

Page 11 line 378
Highly conjugated aromatic hydrocarbons (e.g., anthracene and perylene) or heterocyclic compounds (e.g., phenothiazine) have low Eox1/2 and high excitation energy values, resulting in negative ΔGet (ΔG of electron transfer) values. Therefore, these compounds facilitate photosensitized cationic polymerization when combined with the conventional onium salts. Highly conjugated molecules, thanks to their capability of absorbing light above 400 nm have been shown to be good photoinitiators for the promotion of long-wavelength cationic polymerization [62].

21) L344. correct mechanica,l 
Answer: we have corrected the mistake into the manuscript

22) L343. .... thanks to their excellent mechanica,l thermal and electric conductivity properties. Please explain why fullerenes, graphene and oxidized graphene (graphene oxide) provide these properties.
Answer: We thanks to the referee for this comment and accordingly we have added the following references that explain these properties:
Ref. 
88) Singh, V.; Joung, D.; Zhai, L.; Das, S.; Khondaker, S.I.; Seal, S. Graphene Based Materials: Past, Present and Future. Progress in Materials Science 2011, 56, 1178–1271, doi:10.1016/j.pmatsci.2011.03.003.
89) Das, S.R.; Uz, M.; Ding, S.; Lentner, M.T.; Hondred, J.A.; Cargill, A.A.; Sakaguchi, D.S.; Mallapragada, S.; Claussen, J.C. Electrical Differentiation of Mesenchymal Stem Cells into Schwann-Cell-Like Phenotypes Using Inkjet-Printed Graphene Circuits. Advanced Healthcare Materials 2017, 6, 1601087, doi:10.1002/adhm.201601087.
90) Kausar, A. Fullerene Reinforced Polymeric Nanocomposites for Energy Storage—Status and Prognoses. Frontiers in Materials 2022, 9, doi:10.3389/fmats.2022.874169.
91) Pikhurov, D.V.; Zuev, V.V. The Study of Mechanical and Tribological Performance of Fulleroid Materials Filled PA 6 Composites. Lubricants 2016, 4, doi:10.3390/lubricants4020013.

23) L347 Explain in this manuscript what is acid catalyzed thermal polymerization and how it differs from other types of polymerization. This type of polymerization is used later in the text (section 3.1 and 3.2). The authors assume that the readers are experts in polymerization. Because this is a review, details must be given.
Answer:  In agreement to the referee, we have explained this point as follows:
Page 12 line 417
The nanocomposite was obtained by acid-catalyzed thermal polymerization at 140°C in the presence of a benzyl sulfonium hexafluoroantimonate derivative, which is a thermally latent catalyst. Latent catalysts are inert molecules under normal conditions, i.e., at room temperature, but they show activity by certain external stimuli, such as heating [92,93].

24) L356-357. Include the cationic mechanism of the paper [55]
Answer: In agreement with the referee comment we have included in the manuscript the following figure 6 

Figure 6. Schematic illustration of the in situ cationic polymerization for the preparation of the ESO based paper composites.

25) L365. Include a figure for the structure of xylitol
26) L370. Include a figure for polyurethanes
Answer: we have included the figure 7

Figure 7. Chemical structures of the xylitol and polyurethanes.

27) L374-376. The authors should comment here on the memory-recovery mechanism.
Answer: In agreement with the referee comment we have included in the manuscript the following sentence:
Page 13 line 452
Many polymers show the “shape memory” property that enable them to remember a shape and to return to its original shape after severe deformation in response to certain external stimuli such as temperature, pH, humidity, chemo light or electricity. Among them thermos-responsive shape memory materials are the most interesting thanks to their low cost and high shape recovery ability at relative low temperatures. This property depends on their elastic modulus-temperature behavior, in particular they can be easily deformed ad temperatures above their glass transition temperature (Tg) where they achieve a rubbery elastic state. The shape is then fixed by cooling the material below its Tg and the material can easily return to its original shape by reheating it to a temperature higher than the Tg. [96,97]
Ref. 
96) Tsujimoto, T.; Uyama, H. Full Biobased Polymeric Material from Plant Oil and Poly(Lactic Acid) with a Shape Memory Property. ACS Sustain. Chem. Eng. 2014, 2, 2057–2062, doi:10.1021/sc500310s.
97) Li, F.; Perrenoud, A.; Larock, R.C. Thermophysical and Mechanical Properties of Novel Polymers Prepared by the Cationic Copolymerization of Fish Oils, Styrene and Divinylbenzene. Polymer 2001, 42, 10133–10145.

28) L409-412. The authors at the beginning of the text (possibly in the introduction) should indicate the difference between 3D and 4D printing
Answer: In agreement to the referee comment we have changed the manuscript as follows:
Page 3 line 105 Oil-based biomaterials show also great potential in 4D printing applications. 4D printing is an emerging technique in which the material confers to the 3D printed objects the ability to change form and function for additional capabilities and performance [43,44] In this review the synthesis of the vegetable oil-based biomaterials and their potential applications as scaffolds for regenerative medicine are described in details. 
And we have  added the following references: 
43) Tibbits, S. 4D Printing: Multi-Material Shape Change. Architectural Design 2014, 84, 116–121, doi:10.1002/ad.1710.
44) Ahmed, A.; Arya, S.; Gupta, V.; Furukawa, H.; Khosla, A. 4D Printing: Fundamentals, Materials, Applications and Challenges. Polymer 2021, 228, 123926, doi:10.1016/j.polymer.2021.123926

29) L423-428. Acrylate epoxidized soybean oil (AESO)-based materials (ref. 63) is not used for biomedical applications, but for electronic as well as automotive and aeronautical applications. In addition, acrylates and methacrylates show cytotoxicity (10.1002/(sici)1097-4636(19971215)37:4<517::aid-jbm10>3.0.co;2-5), therefore, it is out of context of biomaterials.
However, the authors could discuss this drawback in terms of structure and use as an (AESO)-based biomaterial. 
Answer: There are many biomaterials made of acrylates and methacrylates polymers that are used as cellular scaffolds in regenerative medicine showing very low cytotoxicity (e.g. PEG-silk fibroin Ciocci I.J.Biomacr. 201, PEG-fibrinogen Seliktar Science  2012; Sil-MA Kim Nature com. 2018 etc). Moreover, we think that AESO-based materials, similarly to ESO-based materials, after functionalization can represent a good biomaterials for biomedical applications, such as described below, see ref 64 (now 107,  Mondal, D. Materials Science and Engineering: C 2021). In this contest we have described the AESO based- biomaterial functionalized with the protein keratin as model of composite AESO based -biomaterial that could also have a potential application in biomedicine for the presence of the keratin protein. 

30) I think in the ref. [64] were evaluated mechanical properties and cytocompatibility of AESO with nano-Hydroxyapatite (nHA) rods and either 2-Hydroxyethyl Acrylate (HEA) or Polyethylene Glycol Diacrylate (PEGDA as function of functional groups. The authors could tackle this interesting point in the review.
Answer: In agreement to the referee comment we have included in the manuscript the following description
Page 16 line 537
In particular, the presence of HEA in the biomaterial significantly increased the rheological properties, extensibility and printability of the nano-composite inks, resulting in homogenous and high-fidelity scaffold structures. However, although the addition of PEGDA lead to improved dispersion, the apparent ultimate compressive strength (UCS), toughness values and elastic modulus were significantly decreased by the addition of either PEGDA or HEA. BM-MSCs showed excellent viability and proliferation on all three types of 3D printed scaffolds. However, the addition of PEGDA in the AESO-based biomaterial also downregulated the osteogenic differentiation of BM-MSCs compared to AESO alone and with HEA. On the contrary, the presence of HEA led to in significant upregulation of osteogenic differentiation of BM-MSCs [107]. 

31) L437-439. The authors could include the reaction used to obtain sustainable and degradable epoxy resins from Trehalose, Cyclodextrin, and Soybean Oil Yield (ref. [65]) and discuss it
Answer: We thanks to the referee for this suggestion. However, the IJMS is an interdisciplinary Journal and this review is also direct to readers with biological formation, so we think that including the reaction in this part of the review could only burden the reading. 

Reviewer 3 Report

This study reviews synthesis and biomedical applications of sustainable vegetal oil-based biomaterials. Waste vegetable oils can represent a good alternative source for the production of biopolymers for several and different applications from the engineering to the biomedicine. It is very important to make an exploration of natural resources as sustainable precursors. I recommend this paper to be published International Journal of Molecular Sciences in present form.

Author Response

 Answer: We thank the referee this positive comment.

Reviewer 4 Report

Comments

The review entitled “Sustainable Vegetal Oil-based biomaterials: Synthesis and Bio-medical Applications”. In this review the authors have tried to focus on the materials generated which are commonly utilized as biomaterials which are well-suited with the environment and inexpensive for biodegradable. The review features the methods for the synthesis of vegetable oil-based biomaterials and the functionalization process to improve the mechanical properties and cell-material interaction for the possible use in regenerative medicines.

However, there are certain areas in this manuscript in which improvement is must needed.

1. Author should include the following references in the introduction section of this review.   

    1) Challenges and opportunities on vegetable oils derived systems for biomedical applications.

        Biomaterials Advances, Volume 134, March 2022, 112720.

    2) Vegetable-oil-based polymers as future polymeric biomaterials.

        Acta Biomaterialia Volume 10, Issue 4, April 2014, Pages 1692-1704.

    3) A Review of Vegetable Oil-Based Polymers: Synthesis and Applications.

       Open Journal of Polymer Chemistry, 2015, 5, 34-40.

    4) Development of Vegetable-Oil-Based Polymers.

        J. APPL. POLYM. SCI. 2014, DOI: 10.1002/APP.40787

2. Please cite the following article in section “Vegetable oils-based biomaterials as scaffolds for          stem cell growth”.

    1) Plant-based biomaterials in tissue engineering.

        Bioprinting Volume 21, March 2021, e00127.

 Decision: Accept after minor revisions.

Author Response

POINT-TO-POINT response to Reviewer’s Comments

First of all, we would like to thank the reviewers for their constructive and positive comments. We hope that our efforts to address the reviewers’ concerns have succeeded in allowing us to prepare a significantly better version of our manuscript. We have devoted a significant time to follow their useful suggestions and address all the comments point-by-point, as shown below.

We hope that the new amended manuscript will meet the requirement of the journal and the interest of the scientific community.

Referee 4

Open Review

English language and style

( ) English very difficult to understand/incomprehensible
( ) Extensive editing of English language and style required
( ) Moderate English changes required
( ) English language and style are fine/minor spell check required
(x) I don't feel qualified to judge about the English language and style

Is the work a significant contribution to the field?

XXX

Is the work well organized and comprehensively described?

XXX

Is the work scientifically sound and not misleading?

XXX

Are there appropriate and adequate references to related and previous work?

XX

Is the English used correct and readable?

XXX

Comments and Suggestions for Authors

Comments

The review entitled “Sustainable Vegetal Oil-based biomaterials: Synthesis and Bio-medical Applications”. In this review the authors have tried to focus on the materials generated which are commonly utilized as biomaterials which are well-suited with the environment and inexpensive for biodegradable. The review features the methods for the synthesis of vegetable oil-based biomaterials and the functionalization process to improve the mechanical properties and cell-material interaction for the possible use in regenerative medicines.

However, there are certain areas in this manuscript in which improvement is must needed.

  1. Author should include the following references in the introduction section of this review.   

1) Challenges and opportunities on vegetable oils derived systems for biomedical applications.

Biomaterials Advances, Volume 134, March 2022, 112720.

2) Vegetable-oil-based polymers as future polymeric biomaterials.Acta Biomaterialia Volume 10, Issue 4, April 2014, Pages 1692-1704.

3) A Review of Vegetable Oil-Based Polymers: Synthesis and Applications.Open Journal of Polymer Chemistry2015, 5, 34-40.

4) Development of Vegetable-Oil-Based Polymers.J. APPL. POLYM. SCI2014, DOI: 10.1002/APP.40787

Answer: we have added the references in the introduction at Line 99 as suggested by the referee 4

31) Ribeiro, A.R.; Silva, S.S.; Reis, R.L. Challenges and Opportunities on Vegetable Oils Derived Systems for Biomedical Applications. Biomaterials Advances 2022, 134, 112720, doi:10.1016/j.msec.2022.112720.

 32) Miao, S.; Wang, P.; Su, Z.; Zhang, S. Vegetable-Oil-Based Polymers as Future Polymeric Biomaterials. Acta Biomaterialia 2014, 10, 1692–1704, doi:10.1016/j.actbio.2013.08.040.

  • Adekunle, K. A Review of Vegetable Oil-Based Polymers: Synthesis and Applications. Open Journal of Polymer Chemistry 2015, 5, 34-40, doi:10.4236/ojpchem.2015.53004.

34)  Islam, M.R.; Beg, M.D.H.; Jamari, S.S. Development of Vegetable-Oil-Based Polymers. Journal of Applied Polymer Science 2014, 131, doi:10.1002/app.40787.

  1. Please cite the following article in section “Vegetable oils-based biomaterials as scaffolds for stem cell growth”.

    1) Plant-based biomaterials in tissue engineering Bioprinting Volume 21, March 2021, e00127.

Answer: we have added the reference at Line 406

  1. Indurkar, A.; Pandit, A.; Jain, R.; Dandekar, P. Plant-Based Biomaterials in Tissue Engineering. Bioprinting 2021, 21, e00127, doi:10.1016/j.bprint.2020.e00127.

Round 2

Reviewer 2 Report

The authors made important improvements to the document, but I need a clean copy to evaluate it. With track changes is very difficult.  Also, reviewing the text and the comments, the following points should be taken into account:

1) The chemical structures in Table 1 and Figure 3 have low resolution. Use the ChemDraw program to improve them. It is a journal with high impact, and well-designed Figures are welcome to the readers.

2) I am not at all convinced of the answer to question 31). I do not see any inconvenient to add a related figure.

3) The response about "shape-memory" is very poor for a review. see: Recent advances in shape–memory polymers: Structure, mechanism, functionality, modeling and applications from Hu et al, . Consider improve this part of manuscript having into account structure of biomaterial polymer and mention, at least, thermodynamic models dealing Young's modulus (10.1088/1361-665X/aaf528).

Author Response

POINT-TO-POINT response to Reviewer’s Comments

First of all, we would like to thank the reviewer 2 for the constructive comments. We hope that our efforts to address the reviewer’ concerns have succeeded in allowing us to prepare a significantly better version of our manuscript. We have devoted a significant time to follow his/her useful suggestions and address all the comments point-by-point, as shown below.

We hope that the new amended manuscript will meet the requirement of the journal and the interest of the scientific community.

Referee 2

Open Review

English language and style

( ) English very difficult to understand/incomprehensible
(x) Extensive editing of English language and style required
( ) Moderate English changes required
( ) English language and style are fine/minor spell check required
( ) I don't feel qualified to judge about the English language and style

Is the work a significant contribution to the field?

X

Is the work well organized and comprehensively described?

XX

Is the work scientifically sound and not misleading?

XX

Are there appropriate and adequate references to related and previous work?

XXX

Is the English used correct and readable?

XXXX

Answer: The manuscript was improved in the English language and style by the suggested MDPI English editing service and we include the English editing certificate  

Comments and Suggestions for Authors

The authors made important improvements to the document, but I need a clean copy to evaluate it. With track changes is very difficult.  Also, reviewing the text and the comments, the following points should be taken into account:

Answer: Most changes were in agreement with the referee 2 ‘s comments and we have submitted the revised version of the manuscript with track changes as requested by the IJMS editorial office. The referee/or the editorial officer could exclude this function showing the manuscript without track changes;

1) The chemical structures in Table 1 and Figure 3 have low resolution. Use the ChemDraw program to improve them. It is a journal with high impact, and well-designed Figures are welcome to the readers.

Answer: Thank you for this comment, in agreement we have provided a high resolution of the table 1 and Figure 3

2) I am not at all convinced of the answer to question 31). I do not see any inconvenient to add a related figure.

Answer: As requested by the referee we have added the figure 9 showing the mechanism described in ref 108 (now 112), add we have included the description in the text

Figure 9. Schematic representation of the CdHS-ESO and TrHS-ESO based composite materials production and the cell scaffolding property of CdHS-ESO. H, heptanoyl chloride; S, succinic anhydride; CdHS, β-cyclodextrin with 6 H units and 8 S units; TrHS, trehalose with 4 H units and 3 S units; nHDFs, neonatal human dermal fibroblasts (adapted from Zhang, Q. et al.) [112].

Page 16 Line 536

        In this study either trehalose (Tr) or β-cyclodextrin (Cd) based materials were produced by reaction with succinic anhydride (S) and heptanoyl chloride (H) in dimethyl formamide and pyridine at 90 °C and then cross-linking with ESO (Figure 9). Both TrHS-ESO and CdHS-ESO were characterized showing that were soft and flexible materials, but that were quickly degraded in basic aqueous conditions. Both the epoxide materials were used as cellular scaffolds, but only the CdHS-ESO material was able to improve the adhesion and proliferation of neonatal human dermal fibroblasts (nHDFs) [112].

3) The response about "shape-memory" is very poor for a review. see: Recent advances in shape–memory polymers: Structure, mechanism, functionality, modeling and applications from Hu et al, . Consider improve this part of manuscript having into account structure of biomaterial polymer and mention, at least, thermodynamic models dealing Young's modulus (10.1088/1361-665X/aaf528).

Answer: We thank to the referee for this comment and for the opportunity to improve our manuscript. In agreement to this comment we have changed the manuscript by addition of the following sentences and refs.:

Page 14 line 445

Extensive knowledge about the structures, functionality, modeling, synthesis methods and applications of shape-memory polymers (SMPs) also in the biomedical field has been achieved [98,99]. Moreover, the multi-shape memory effect (multi-SME), that is the property of polymers to memorize more than two temporary shapes [100], can be obtained by amorphous SMPs with hard and soft segments, significantly improving the SMPs potential and practical applications [101]. In this context, a thermodynamic model to explain the working mechanism and thermomechanical behavior of SMPs, with tunable multi-SME, has been developed by Lu et al. [101].

Page 17  Line 556

Castor oil (CO), which is a vegetable-oil obtained from the Ricinus communis and characterized by the presence of triglycerides with hydroxylated groups, has been also used as chain extender for the production of SMPs [115]. CO, used as chain extender in PUs synthesis by polytetramethylene glycol (PTMG250), was able to improve the shape memory capacity of the PU, increasing its Tg and biocompatibility. The epithelial cells (HEK293 cell line) seeded on this SMP, after 72 h of cell growth, showed an adherent spindle shape and normal morphology by microscopic analyses [115], demonstrating that this biomaterial was suitable for applications in biomedicine.

Page  17 line 568

Due to their shape memory property EVO-based SMPs can have application in biomedical field not only as scaffolds for tissue engineering but also as drug delivery systems and medical devices for minimally invasive surgery, i.e. removable intravascular and ureter stents, degradable sutures and orthodontic tools.

Added References  

98) Hu, J.; Zhu, Y.; Huang, H.; Lu, J. Recent Advances in Shape–Memory Polymers: Structure, Mechanism, Functionality, Modeling and Applications. Progress in Polymer Science 2012, 37, 1720–1763, doi:10.1016/j.progpolymsci.2012.06.001.

99)Serrano, M.C.; Ameer, G.A. Recent Insights Into the Biomedical Applications of Shape-Memory Polymers. Macromolecular Bioscience 2012, 12, 1156–1171, doi:10.1002/mabi.201200097.

100) Xie, T. Tunable Polymer Multi-Shape Memory Effect. Nature 2010, 464, 267–270, doi:10.1038/nature08863.

101) Lu, H.; Wang, X.; Yu, K.; Fu, Y.Q.; Leng, J. A Thermodynamic Model for Tunable Multi-Shape Memory Effect and Cooperative Relaxation in Amorphous Polymers. Smart Materials and Structures 2019, 28, 025031.

116) Veloso-Fernández, A.; Laza, J.M.; Ruiz-Rubio, L.; Martín, A.; Taguado, M.; Benito-Vicente, A.; Martín, C.; Vilas, J.L. Towards a New Generation of Non-Cytotoxic Shape Memory Thermoplastic Polyurethanes for Biomedical Applications. Materials Today Communications 2022, 33, 104730, doi:10.1016/j.mtcomm.2022.104730.
